

# WRF v.3.9 sensitivity to land surface model and horizontal resolution changes over North America

Almudena García-García[1, 2], Francisco José Cuesta-Valero[1], Hugo Beltrami[1],
Fidel González-Rouco[3], and Elena García-Bustamante[4]

[1]Climate & Atmospheric Sciences Institute, St. Francis Xavier University, Antigonish, Nova Scotia, Canada
[2] Department of Remote Sensing, Helmholtz Centre for Environmental Research – UFZ, Leipzig, Germany
[3]Physics of the Earth and Astrophysics Department, IGEO (UCM-CSIC), Universidad Complutense de Madrid, Spain.
[4]Research Center for Energy, Environment and Technology (CIEMAT), Madrid, Spain.

**Correspondence:** Hugo Beltrami (hugo@stfx.ca)

**Abstract.** Understanding the differences between regional simulations of land-atmosphere interactions and near-surface conditions is crucial for a more reliable representation of past and future climate. Here, we explore the effect of changes in the model's horizontal resolution on the simulated energy balance at the surface and near-surface conditions using the Weather Research and Forecast-

ing (WRF) model. To this aim, an ensemble of twelve simulations using three different horizontal resolutions (25 km, 50 km and 100 km) and four different Land Surface Model (LSM) configurations over North America from 1980 to 2013 is developed. Our results show that finer resolutions lead to higher surface net shortwave radiation and maximum temperatures at mid- and high latitudes. At low latitudes over coastal areas, an increase in resolution leads to lower values of sensible

heat flux and higher values of latent heat flux, as well as lower values of surface temperatures and higher values of precipitation and soil moisture in summer. The use of finer resolutions leads then to an increase in summer values of latent heat flux, convective and non-convective precipitation and soil moisture at low latitudes. The effect of the LSM choice is larger than the effect of horizontal resolution on the near-surface temperature conditions. By contrast, the effect of the LSM choice

on the simulation of precipitation is weaker than the effect of horizontal resolution, showing larger differences among LSM simulations in summer and over regions with high latent heat flux. Comparison between observations and the simulation of daily maximum and minimum temperatures and accumulated precipitation indicates that the CLM4 LSM yields the lowest biases in maximum and minimum mean temperatures, but the highest biases in extreme temperatures. Increasing horizontal

resolution leads to larger biases in accumulated precipitation over all regions particularly in summer. The reasons behind relate the partition between convective and non-convective precipitation, specially noticeable over western US.





## 1 Introduction

Most assessments of climate change impacts on ecosystems and societies are based on projections
performed by Regional Climate Models (RCMs) and/or Earth System Models (ESMs, IPCC, 2013;
Arneth, 2019). Exploring inter-model differences in present climate simulations is necessary to un-
derstand their contribution to the spread in future climate projections, and ultimately to better char-
acterize or even reduce the uncertainty in the simulation of the response to a given scenario (Cubasch
et al., 2013). Understanding inter-model differences is also important for paleoclimatic studies re-
lying on regional climate model simulations to bridge the gap between the local character of proxy
reconstructions and ESM global simulations (e.g. PALEOLINK, Gómez-Navarro et al., 2018).

The representation of land-atmosphere interactions within climate models has received consider-
able attention during the last decades due to their influence on surface climate, vegetation and soil
hydrology, and therefore, on climate variability (e.g. Lorenz et al., 2016; Vogel et al., 2017). For ex-
ample, energy and water exchanges between the lower atmosphere and the ground surface have been
shown to alter surface conditions, particularly during extreme weather events in summer (Senevi-
ratne et al., 2006; Hirschi et al., 2011; Miralles et al., 2012; Hauser et al., 2016). Land-atmosphere
interactions have also been studied in the evaluation of climate model simulations, applying sev-
eral metrics to characterize surface energy fluxes and near-surface conditions (Koven et al., 2013;
Dirmeyer et al., 2013; Sippel et al., 2017; García-García et al., 2019).

The representation of near-surface conditions (e.g. air and soil temperatures, soil moisture...) and
energy and water exchanges at the land surface in a climate model depends on the processes sim-
ulated by the atmospheric and soil model components, and on the degree of coupling implemented
between both model components (Koster et al., 2006; Melo-Aguilar et al., 2018). Different Land
Surface Models (LSMs) include varying levels of realism in the representation of soil physics. Thus,
each LSM simulates somewhat different surface water and energy fluxes (e.g. Lawrence et al., 2019).
For example, the representation of surface albedo, evaporative resistance and aerodynamic rough-
ness by each LSM alters the simulation of the surface energy balance and consequently affects the
evolution of surface temperatures (Laguë et al., 2019; MacDougall and Beltrami, 2017). The depen-
dence of the simulated land-atmosphere interactions on the LSM has been shown in previous studies
using global (García-García et al., 2019), and regional climate model simulations over North Amer-
ica (Pei et al., 2014; García-García et al., 2020). There are examples of these studies at local scales
(Mooney et al., 2013; Wharton et al., 2013; Chen et al., 2014; Van Den Broeke et al., 2018; Liu
et al., 2019; Zhuo et al., 2019) and at continental scales for Europe (Davin and Seneviratne, 2012;
Mooney et al., 2013).





Horizontal resolution is another factor to take into account in the study of land-atmosphere interactions in climate model simulations. ESMs are limited by computational resources, using horizontal resolutions from approximately 250 to 100 km (e.g. CMIP5 models, Taylor et al., 2012), while RCMs allow for using much finer resolutions. Analogously, the range of horizontal resolu-

tions employed in RCMs for climate studies is usually limited to approximately 50-25 km (e.g. CORDEX models, Giorgi and Gutowski Jr., 2015). In spite of this enhanced resolution, RCM ability to reproduce temporal variability like that of precipitation at daily time scales is still limited, being greatly improved by using resolutions of $\sim 4$ km and convection resolving RCMs (Sun et al., 2016). Previous studies have shown some resolution-induced improvements in the simulation of

precipitation, wind and high altitude temperatures at local and regional scales with possible implications for the simulation of climate dynamics (Ban et al., 2014; Gómez-Navarro et al., 2015; Messmer et al., 2017; Hahmann et al., 2020; Vegas-Cañas et al., 2020). Small scale weather phenomena such as sea breezes, snowstorms induced by the presence of lakes, local winds, tropical cyclones, and mesoscale convective systems can be better represented with increased resolution (Wehner et al.,

2010). Some studies have also suggested a resolution-induced improvement in the representation of interactions between small and large scale dynamical processes, ultimately leading to better large-scale atmospheric flow (Lucas-Picher et al., 2017). Thus, the difference in the resolution employed in RCMs and GCMs is expected to improve the representation of land-atmosphere interactions in RCMs through a more adequate discretization of equations, as well as through a more realistic repre-

sentation of small-scale processes and topographical features (Xue et al., 2014; Rummukainen, 2016; Vegas-Cañas et al., 2020). The improved representation of land-atmosphere interactions associated with finer resolutions is also expected to induce an improvement in the simulation of near-surface conditions, especially in the simulation of extreme events (Prein et al., 2013; Di Luca et al., 2015; Rummukainen, 2016; Demory et al., 2014).

Although the literature on the impact of the LSM choice and changes in resolution on climate simulations is extensive, most studies are focused on small domains and meteorological events, providing little information about the impact of the LSM choice and horizontal resolution changes on long-term climatological variability. Here, we evaluate and compare the influence of both factors, the LSM choice and horizontal resolution, on the representation of energy and water fluxes at the

surface and consequently on the simulation of near-surface conditions over North America for a climatological period spanning the time interval 1980-2013. We use the WRF model (version 3.9, Michalakes et al., 2001), that allows for testing a number of LSM schemes, each varying in physical parameterization complexities. An ensemble of twelve simulations was generated using three different horizontal resolutions (25 km, 50 km and 100 km), four different LSM configurations and two

vegetation options, i.e. prescribed or dynamical vegetation.





The WRF representation of climate at different resolutions is expected to affect the simulation of atmospheric and surface phenomena in several ways, through different representation of cloud formation, or through different level of details in the description of orography and land cover. Simulations with different LSMs provide information about the advantages of using a comprehensive

LSM, such as the CLM4 LSM (Oleson et al., 2010), or a simpler LSM component, such as the NOAH LSM (Tewari et al., 2004). Additionally, the comparison of two identical model configurations except for the prescribed or dynamical vegetation mode provides information about the effect of a realistic evolution of vegetation cover on the energy and water balance at the surface. Thus, soil and near-surface variables from these twelve simulations are compared and also evaluated against

temperature and precipitation observations to explore two questions: 1) How do changes in horizontal resolution within WRF affect the simulation of land-atmosphere interactions and near-surface conditions? 2) How do LSM differences in the simulation of land-atmosphere interactions translate into different near-surface conditions?

The descriptions of the WRF experiments and the methodology applied for the analysis are in-

cluded in Section 2 and 3, respectively. Section 4 presents the results of the analysis, which are discussed using the available literature in Section 5. The conclusions and importance of this work are summarized in Section 6.

## 2  Description of the modelling experiment

We performed three sets of regional simulations (twelve simulations in total) over North America

from 1980 to 2013 using the Advanced Research WRF model (WRF v3.9, Michalakes et al., 2001) with initial and boundary conditions from the North American Regional Reanalysis product (NARR, Mesinger et al., 2006). The NARR product was generated by the NCEP Eta atmospheric model (ZI, 1997), the NOAH LSM component (Mitchell, 2005) and the Regional Data Assimilation System (RDAS Mesinger et al., 2006). The NARR data were obtained from the National Center for Envi-

ronmental Information (NOAA) archive and provides data over a 32 km grid with a three-hourly temporal resolution. No nudging techniques were applied within the domain of the simulation. The use of nudging techniques imposes large scale variability within the inner scales of the domain of simulation, thus partially muting the generation of local and regional scale dynamical responses in favour of representing the large scale flow of the driving conditions. Avoiding nudging, therefore,

allows for more clearly expressing the influences of increased resolution and of changing the LSM component (von Storch et al., 2000).

The three sets of simulations were performed using a Lambert type projection with resolutions of $25\times25$ km, $50\times50$ km, and $100\times100$ km. Each set includes four simulations using three different





LSM components: the NOAH LSM (NOAH, Tewari et al., 2004), the NOAH LSM with multiparam-
eterizations options (NOAH-MP, Niu et al., 2011), and the Community Land Model version 4 LSM
(CLM4, Oleson et al., 2010). The fourth simulation included in each set was performed using the
NOAH-MP LSM with dynamic vegetation (NOAH-MP-DV), while vegetation was prescribed for
the other simulations. The rest of the WRF options remained the same for all simulations, employ-
ing 27 atmospheric levels, land cover categories from the Moderate Resolution Imaging Spectrora-
diometer (MODIS, Barlage et al., 2005), the WRF Single Moment (WSM) 6-class graupel scheme
for the microphysics (Hong and Lim, 2006), the Grell-Freitas ensemble scheme (Grell and Freitas,
2014), the Yonsei University scheme for the description of the planetary boundary layer scheme
(YSU, Hong et al., 2006), the revised MM5 monin-Obukhov scheme (Jiménez et al., 2012), and the
Community Atmosphere Model (CAM) scheme (Collins et al., 2004). The use of different horizon-
tal resolutions requires the use of different time steps for performing our WRF simulations, as well
as different time intervals for computing radiation physics (radt option in WRF namelist). Table 1
summarizes the differences between all simulations employed in this analysis.

The LSM schemes used in this study differ in parameterization complexity, being the NOAH the
most basic amongst the three selected LSMs. The NOAH LSM describes soil and vegetation pro-
cesses for the closure of the water and energy budgets, discretizing the soil into four soil layers that
reach a total of 2m of depth (Tewari et al., 2004). Some limitations have been associated in this soil
scheme with the represented bulk layer of canopy, snow and soil, its system to drain water at the bot-
tom of the soil column and its simple snow melt dynamics (Wharton et al., 2013). The NOAH-MP
version of the NOAH LSM improves soil hydrology and the representation of terrestrial biophysical
processes (Niu et al., 2011). This scheme includes a separate vegetation canopy with a compre-
hensive description of vegetation properties. The NOAH-MP also includes a multi-layer snow pack
description with liquid water storage and melting and refreezing capabilities. It includes the same
soil structure than the NOAH LSM (four soil layers, down to 2 m).The CLM4 LSM incorporates a
comprehensive representation of biogeophysics and hydrology, including a single-layer vegetation
canopy, a five-layer snowpack, and a ten-layer soil column down to a depth of 3.802 m. This scheme
also characterizes each grid cell into five primary sub-grid land cover types (glacier, lake, wetland,
urban, and vegetated), using up to 4 plant functional types (PFTs) to describe vegetation physiology
and structure.

## 3 Methodology

We study the impact of changing horizontal resolution on surface energy fluxes and near-surface
conditions as simulated by the WRF model with different LSM components. For this purpose, we



estimated the temporal averages of surface energy fluxes for the analysis period (1980-2013) focusing on the following energy components: net shortwave radiation (SNET, $W/m^2$), net longwave radiation (LNET, $W/m^2$), net radiation absorbed by the soil (RNET, $W/m^2$), latent heat flux (LH, $W/m^2$), sensible heat flux (HFX, $W/m^2$) and ground heat flux (GHF, $W/m^2$). The temporal averages of near-surface conditions are estimated using outputs of 2 m air temperatures (SAT, $^oC$), daily maximum SAT (TASMAX, $^oC$), daily minimum SAT (TASMIN, $^oC$), soil temperature at 1m depth (GST 1m, $^oC$), accumulated convective and non-convective precipitation at the surface (RAIN C and RAIN NC, $mm/day$), soil moisture contained in the first soil meter (SM 1m, $m^3/m^3$) and total cloud fraction (TCLDFR, %). All values are computed using the annual and seasonal (boreal winter, DJF; summer, JJA) averages over the 34-year period (1980-2013) after discarding the first year of the simulation (1979) as spin up, which is enough to avoid the effect of initial conditions (García-García et al., 2020). Thus, we estimated the anomalies of all outputs for each LSM simulation relative to the multi-model mean (the mean of CLM4, NOAH, NOAH-MP, and NOAH-MP-DV outputs) for each set of simulations with different resolution (25 km, 50 km and 100km). Similarly, we estimated the change in the simulation of all variables between the 100 km and 50 km simulations and between the 50 km and 25 km simulations for all LSM configurations. When required, outputs of all WRF experiments were mapped to the grid of the observational reference employed in this study by selecting the nearest model grid point. A Student's t-test considering autocorrelation was used to identify significant differences between simulations with different LSMs and horizontal resolutions at the 95% confidence level.

Additionally, we evaluate the WRF performance in the simulation of maximum and minimum air temperature and accumulated precipitation against the Climatic Research Unit Time-Series product version 4.03 (CRU, Harris et al., 2014). The CRU database provides monthly data at a resolution of approximately 50 km. Previous studies have reported inconsistencies between different observational products, particularly important for the evaluation of model simulations with different resolutions (e.g. Iles et al., 2019). To avoid this issue, we included another observational database in the analysis; the Daily Surface Weather Data version 3 (DAYMET Thornton et al., 2016), with daily data at approximately 1 km resolution over North America. This allows the comparison of uncertainties associated with the choice of the observational product and the uncertainties arising from the model configuration. We calculated annual and seasonal WRF and DAYMET biases in these variables relative to the CRU data for the analysis period averaging over six subregions, due to the large climate differences over North America. These six subregions cover Central and North America (NA) and are adapted from Giorgi and Francisco (2000): Central America, CAM; Western North America, WNA; Central North America, CNA; Eastern North America, ENA; Alaska, ALA; and Greenland, GRL (Figure 1). The impact of horizontal resolution is expected to be larger on the





simulation of extreme events than on surface climatologies (Prein et al., 2013; Di Luca et al., 2015; Rummukainen, 2016). We examined this by calculating the bias in the 95th percentile of maximum and minimum temperature and accumulated precipitation within all experiments using the DAYMET

product as reference.

## 4  Results

### 4.1  LSM influences on surface energy fluxes and near-surface conditions

The net radiation absorbed by the ground surface enhances turbulent (latent and sensible) fluxes at the surface and/or warms the soil surface, which leads to an increase in the emitted longwave

radiation (Bonan, 2002). The relationship between these variables is shown by their corresponding ensemble mean of LSM simulations, indicating similar latitudinal patterns in net radiation, turbulent fluxes and near-surface temperatures with higher fluxes and temperatures at lower latitudes (see for example Figure 2 for the LSM ensemble mean of the 50 km experiments). The net radiation results from adding net shortwave and longwave radiations, whose mean values have similar spatial

distributions but with opposite sign (Figure 2a). This indicates that more shortwave radiation reaches the land surface than is reflected due to surface albedo, while the radiation emitted from the soil due to its surface temperature, is higher than the longwave radiation reaching the soil surface (Figure 2a). The energy proportion of net radiation that is propagated through the soil is much smaller than the rest of surface energy fluxes (Figure 2b). Areas with high latent heat flux coincide with areas with

high convective precipitation rates at low and middle latitudes (Figure 2b and d). The soil moisture map shows high values southward the Great Lakes region and low values in dry areas such as Florida and the southwest US, in agreement with both convective and non-convective precipitation (Figure 2d and e).

The use of a different LSM component in WRF affects the representation of soil and vegetation

properties and processes in the simulation, resulting in noticeable differences in the simulated energy fluxes across the domain (Figure 3). For instance, the spatial patter of the LSM anomalies relative to the multi-model mean in longwave net radiation is similar to the minimum temperature anomalies (Figures 3b and 4b), and the anomalies in latent heat flux show similar LSM differences to the convective precipitation maps at low latitudes (Figures 3d and 5a). The spatial patter of the LSM

anomalies in sensible and latent heat fluxes shows opposite values around the mean. For example over the boreal forest, the LSMs reaching the highest values of latent heat flux also reaches the lowest values of sensible heat flux at the same locations (Figure 3d and f).

LSM differences in the simulation of surface fluxes and near-surface conditions are similar among the experiments with different resolutions (Figures 3, 4, 5, S2, S4 and S6). For example, based on



the differences between each LSM simulation with 50 km resolution and the 50 km multi-model
      mean (Figure 2), we can identify the CLM4 as the LSM component simulating the highest net short-
      wave radiation over most of North America for the annual, DJF and JJA means (SNET in Figures
      3a and S1). Meanwhile, the NOAH-MP-DV simulation reaches the lowest net shortwave radiation
      over the same areas (Figure 3a). The WRF simulation of net longwave radiation reaches negative

values (Figure 2a) with the maximum values simulated by the NOAH LSM and the minimum values
      simulated by the CLM4 LSM (Figure 3b). The upward (negative) component of the net longwave
      radiation results from the Stefan-Boltzmann equation, from which the out-going long-wave radia-
      tion $LW \uparrow$ is proportional to $\sigma T_s^4$, where $T_s$ is surface temperature and $\sigma$ is the Stefan-Boltzmann
      constant. Thus, the CLM4 simulation produces the largest outgoing longwave radiation (see Figure

S7), shown in Figure 3b by the largest negative anomalies and therefore the highest air and soil tem-
      peratures (Figure 4). The opposite behaviour is performed by the NOAH LSM, yielding the lowest
      upward longwave radiation (Figures 3b and S7), and one of the coldest temperature climatologies
      relative to the multi-model mean (Figure 4). The relationship between the radiation and temperature
      anomalies is also supported by high spatial correlation coefficients (Table S1). These correlation

coefficients show the link of maximum temperatures to both shortwave and longwave net radiation,
      particularly in summer, while minimum temperatures show higher correlation with longwave net
      radiation than with shortwave net radiations in most of LSM simulations (Table S1).

      The simulation of sensible heat flux reaches the highest values using the CLM4 configuration over
      the boreal forest and the lowest values in western US. Meanwhile, the NOAH simulation reaches the

lowest sensible heat fluxes over the boreal forest and the highest values in western US (Figure 3e).
      The spatial patterns of LSM anomalies in sensible heat flux are similar to the LSM anomalies in net
      shortwave radiation and daily maximum temperatures (Figures 3a and 4a), which is also shown by
      high spatial correlation coefficients, particularly in summer (Table S1). LSM differences in ground
      heat flux are smaller than for the rest of the energy fluxes due to the small magnitude of the GHF

in comparison with the rest of energy components (Figure 2g). The NOAH LSM reaches the lowest
      ground heat flux values for the annual mean in most of the domain, showing nonetheless the highest
      ground heat flux values in summer at high latitudes (Figures 3f and S1). The spatial pattern of LSM
      differences in ground heat flux differ from the soil temperature results, whose spatial correlation
      coefficients are higher with the longwave net radiation mainly in summer. LSM differences are

larger for the simulation of soil temperatures than for the simulation of air temperatures particularly
      at high latitudes in summer where LSM differs largely in the simulation of shortwave net radiation,
      probably due to different estimates of surface albedo under different land cover and soil moisture
      values (Figures 2a, 4c-d, 5c, and S3).





The CLM4 simulation produces the highest latent heat flux values and convective precipitation

rates, particularly over southwestern NA, while the NOAH simulation provides the lowest latent heat flux and convective precipitation values over the same areas (Figures 3d and 5a). LSM differences in latent heat flux, convective and non-convective precipitation are larger in summer (Figures S1 and S5). LSM differences in JJA convective precipitation rates are particularly large at low latitudes, where the CLM4 LSM produces high latent heat flux over the western part of the domain

and low latent heat flux over the eastern NA relative to the multi-model mean (Figure S1 and S5). This differences between the east and west in the CLM4 simulation of latent heat flux are not reflected in the values of convective precipitation rates, hence the low spatial correlation coefficients between both variables (Table S2). There are also LSM differences in the non-convective term of precipitation, with larger anomalies in summer than in winter (Figures 5b and S5). Thus, the NOAH

LSM produces the highest precipitation anomaly at mid-latitudes, where the same LSM produces high values of total cloud fraction relative to the multi-model mean (Figure 5b and d). This relationship between non-convective precipitation and cloud cover is also shown by high spatial correlation coefficients (Table S2). There are large LSM differences in the simulation of soil moisture, with the CLM4 LSM generating the driest conditions and the NOAH-MP experiments the wettest soils (Fig-

ures 5c and S5). Although the LSM differences are large in the representation of humid conditions, the relationship between precipitation, latent heat flux, soil moisture and cloud fraction is weaker than the relationship between radiation and temperatures in the WRF experiments.

    Larger LSM differences in the simulation of shortwave and longwave radiations and temperatures are found over very vegetated areas such as the boreal forest and eastern US (Figures 3 and

4). The comparison of the NOAH-MP experiments with prescribed and dynamic vegetation also show larger differences in air temperatures over these areas. This suggests that the different representation of vegetation in each LSM yields to different estimates of soil properties, such as surface albedo, evaporative resistance and aerodynamic roughness, affecting the simulation of radiation and temperatures. The LSM differences in latent heat flux and convective precipitation are larger at low

latitudes, with the NOAH LSM yielding the most different values relative to the rest of LSMs. This can be associated with the issues identified in the NOAH LSM to simulate soil hydrology (Wharton et al., 2013).

## 4.2   Resolution impact on surface energy fluxes and near-surface conditions

The response of surface energy fluxes and near-surface conditions to changes in spatial resolution

varies considerably with the season, while they behave similarly using different LSM components (Figures 6, S8 - S13). We consider the results of the NOAH-MP-DV experiments as example to minimize the number of figures in the manuscript. Changes in resolution alter the surface energy fluxes





in DJF mainly over regions of complex topography and coastal areas, while in JJA the simulation of energy fluxes is affected by resolution over the whole domain except over a region in the central
US and northern areas of Hudson Bay (Figure 6). In JJA, the use of coarser horizontal resolutions induces a decrease in the net radiation absorbed by soil at high latitudes, mainly caused by a decrease in net shortwave radiation (Figure 6a and c). The decrease in net radiation induced by coarser resolutions limits the energy available for turbulent energy fluxes at high latitudes. Thus, values of latent and sensible heat fluxes also decrease at coarser resolutions at high latitudes (Figure 6d and
e). The use of coarser resolutions also induces higher sensible heat fluxes at low latitudes, which is balanced by lower latent heat flux (Figure 6d and e).

Consistently with the effect of LSM differences on near-surface conditions (Section 4.1), the spatial pattern of the resolution impact on net shortwave radiation is similar to the resolution-induced changes in daily maximum temperatures (Figures 6a and 7a). The response of minimum tempera-
ture to changing resolution is, however, smaller than for maximum temperatures at mid-high latitudes (Figure 7b). Over eastern North America, JJA minimum temperature increases with the use of coarser resolutions, while it decreases over western North America (Figure 7b). The response of mean temperature to resolution is mainly driven by the resolution impact on maximum temperatures, since both show similar spatial patterns (Figure 7c). Although the impact of resolution on air
and soil temperature shows similar spatial patterns, in winter air temperatures are more affected by changes in resolution than soil temperatures, particularly at high latitudes and elevation where snow may be present (Figure 7c and d). Thus, soil temperatures are more sensible to JJA changes in the energy budget induced by reducing resolution, while in DJF soil temperature remains insulated from resolution-induced changes in surface conditions, probably because of the insulating effect of snow
cover (García-García et al., 2019).

Non convective precipitation increases with the use of coarser horizontal resolutions over the Rocky Mountains, particularly in winter, where the model also represents higher percentage of cloud cover with coarser resolutions (Figure 8b and d). Over the east coast of the US, the use of coarser resolutions leads to lower non-convective precipitation rates (Figure 8b). This behaviour is also
present in JJA over the arctic areas of our domain (Figure 8b). Although the response of convective precipitation to resolution in winter is not significant, in JJA the use of coarser horizontal resolutions yields a decrease in convective precipitation over coastal areas and the Rocky Mountains, where the simulation also reaches lower latent heat flux values (Figures 6d and 8a). Although the spatial pattern of soil moisture is very patchy in DJF and JJA, soil moisture tends to decrease at low latitudes in
JJA with the use of coarser resolutions (Figure 8c). At mid-latitudes, however, the use of coarser resolution leads to an increase in soil moisture during the year at most locations (Figure 8c).





In summary, at low NA latitudes the JJA values of the three variables associated with the surface water balance (LH, Rain and SM 1m) decrease with the use of coarser horizontal resolutions, although showing large spatial variability (Figures 6d and 8a and c). At mid- and high NA latitudes, there are differences in the response of the water balance variables to the use of coarser resolutions. For example, soil moisture increases with coarser resolutions over a large area at mid-latitudes, while convective precipitation increases just over a few grid cells in central NA, decreasing over most coastal areas (Figure 8a and c). Latent heat flux decreases with the use of coarser resolutions over most regions at high and mid-latitudes, particularly over coastal areas (Figure 6d).

### 4.3 Comparison of temperature and precipitation against observations

For the comparison of the effect of LSM and horizontal resolution changes on climate simulations, we estimate the bias in all WRF simulations relative to the CRU observational product (Harris et al., 2014). As a measure of the possible uncertainties in observational databases, we also estimate the bias in the DAYMET product relative to the CRU data. The inconsistencies between both observational data are noticeably smaller than the biases in the WRF experiments for all variables in all regions, except for minimum temperatures in the CAM region and for precipitation in the ALA region (Figure 9, see a representation of the regions in Figure 1).

The WRF model underestimates annual and seasonal means of daily maximum temperatures over most of North America at all resolutions, comparing grid-cell by grid-cell (Figure S14) and on average over subdomains (Figure 9a). These biases are generally less pronounced for the experiments using the CLM4 LSM at most locations and in all seasons. The impact of horizontal resolution on these values is weaker than the LSM dependence over each domain, showing a larger effect of resolution on maximum temperatures in summer than in winter over western North America and at high latitudes (WNA, ALA and GRL in Figure 9a). Over these areas, finer horizontal resolutions are associated with warmer JJA maximum temperatures, reducing the bias relative to the CRU dataset at middle latitudes (Figures 7a and 9a). In the ALA and GRL regions, the WRF model with the CLM4 and the NOAH-MP LSM components overestimates JJA maximum temperatures, increasing the bias in these simulations with the use of finer resolution (Figure 9a). The WRF bias in maximum temperatures in winter is greatly improved over the boreal forest and the Rocky areas by using the CLM4 as soil scheme (Figure S14). Over the same areas the CLM4 simulated very high values of shortwave net radiation and sensible heat flux in comparison with the rest of land surface models (Figure 3a and e), which may be related to the CLM4 albedo estimate. Despite the WRF underestimation of mean maximum temperature, extreme maximum temperatures are overestimated by the WRF model, particularly at high latitudes and using the CLM4 LSM (Figure 10a). As expected based on the literature, the resolution effect on the bias in extreme maximum temperatures is larger





than on the bias in mean maximum temperatures, however LSM differences are still larger than resolution-induced changes.

The performance of the WRF model in reproducing daily minimum temperatures from the CRU observations is slightly better than reproducing maximum temperatures at mid- and low latitudes,

but it is worse at high latitudes particularly in DJF (Figure 9b). Experiments using the CLM4 LSM yield a warmer climatology over most areas and for all seasons than the experiments with the other LSM components, implying smaller biases in the CLM4 simulations for most of regions (Figure 9b). The WRF bias in minimum temperature is large in winter over the central and eastern areas of North America and at mid- and high latitudes (subdomains ALA, GRL, CNA and ENA in Figure

9b). The resolution impact on these results is again weaker than the effect of the LSM component. In summer, the WRF-NOAH experiments show a large negative bias in minimum temperatures over the NA southeastern coast (Figure S15), areas where the same experiments also shown high values of longwave net radiation and low values of sensible heat flux (Figure 3b and e). The simulation of extreme minimum temperatures is also overestimated for all experiments and regions except for the

CAM region in summer, showing particularly large biases in winter (Figure 10b). Thus, although the WRF model underestimates mean maximum and minimum temperatures, it overestimates the intensity of hot extremes associated with minimum and maximum temperatures. The effect of dynamic vegetation (NOAH-MP vs NOAH-MP-DV) on the biases in extreme and mean maximum and minimum temperatures remains constant using different resolutions, reaching larger biases and colder

maximum and minimum temperatures with dynamic vegetation than with prescribed vegetation for most of the regions (Figure 9a and b).

The WRF model simulates large positive biases in daily accumulated precipitation at the surface over most of North America during all seasons, with larger biases in summer (Figures 9c and S16). A negative bias is also present in all experiments over the southeastern US and the eastern coast

of North America in summer and winter (Figure S16). Dry biases are reduced when using finer horizontal resolutions, while wet biases are larger when using smaller scales (Figure 9c). This is due to the intensification of the water cycle with the use of finer horizontal resolutions discussed in section 4.2 and presented in Figures 6d and 8. For example in winter, the dry bias shown in all experiments over the southeastern NA is associated with an increase in non-convective precipitation

using finer resolutions (Figures S16 and 8b). In summer, the bias in precipitation is larger using finer resolutions over most of coastal areas where an increase in convective precipitation and latent heat flux were shown with the use of finer resolutions (Figures 6d, 8a and S16). The impact of resolution on the accumulated precipitation is stronger than the effect of the LSM component, which affects precipitation mainly in summer (Figure 9). The results show larger bias in extreme precipitation than

for the mean accumulated precipitation, but yielding similar conclusions (Figure 10).





In summary, the LSM impact on temperatures is larger than the resolution effect, while the opposite is true for precipitation climatologies, i.e. differences in precipitation arising from changes in resolution are larger than LSM differences (Figure 9). The influence of both the LSM choice and resolution intensifies in summer comparing with the rest of seasons, probably because of the larger energy exchanges and the consequent intensification of land-atmosphere coupling in summer (Zhang et al., 2008; Mei and Wang, 2012). The CLM4 LSM generates the smallest biases relative to the CRU database in the WRF simulation of mean maximum and minimum temperature, however it also yields the larger biases in extreme maximum and minimum temperatures. The use of finer resolutions leads to slightly larger or smaller biases in the simulation of maximum and minimum temperatures depending on the LSM component and the region, while the use of finer resolutions implies larger biases in mean and extreme precipitation at low and mid-latitudes for all LSM components, particularly in summer.

## 5 Discussion

The dependence of climate simulations on the LSM component shown in this study agrees with conclusions drawn from previous analyses at different temporal and spatial scales (Chen et al., 2014; Van Den Broeke et al., 2018; Liu et al., 2019; Zhuo et al., 2019; Davin and Seneviratne, 2012; Mooney et al., 2013; Lagüe et al., 2019; García-García et al., 2020). For example, using the Consortium for Small-scale Modeling (COSMO) and WRF RCMs, Davin and Seneviratne (2012) and Mooney et al. (2013) identified a LSM sensitivity of temperature and precipitation conditions over Europe, which intensifies in summer. Additionally, our analysis has shown that the impact of the LSM choice on the WRF simulation of precipitation is weaker than its impact on temperature, in agreement with studies over a small region in Italy (Zhuo et al., 2019), over Europe (Mooney et al., 2013) and over the western and central US at seasonal scales (Jin et al., 2010; Chen et al., 2014; Van Den Broeke et al., 2018).

The large WRF sensibility of precipitation rates to resolution is also supported by the literature (e.g. Pieri et al., 2015). Our results also shown large seasonal differences, mainly caused by the different contribution of convective precipitation in summer and winter. In summer at middle and low latitudes, the use of finer grid cells leads to a change in the energy partition into sensible and latent heat flux, increasing latent heat and decreasing sensible heat (Figure 6). This increase in latent heat flux over these areas is probably the caused of the higher values of convective precipitation (Figure 8a). This resolution-induced increase in precipitation through changes in convective processes has also been suggested in the literature (Prein et al., 2016). At high latitudes both turbulent heat fluxes





increase in summer with the use of finer resolutions, mainly due to the increase in shortwave net radiation probably related to the decrease in cloud cover shown in Figure 8d.

Previous evaluations of RCMs using several soil schemes over different domains reached the conclusion that the most complex LSM components, that is, the LSM components representing more physical phenomena, outperform others (Chen et al., 2014; Van Den Broeke et al., 2018; Liu et al., 2019). Over North America, our results indicate that the WRF simulation of temperature conditions using the CLM4 LSM outperforms the simulation of mean maximum and minimum temperatures

generated by the NOAH and NOAH-MP LSMs, but it yields larger biases in extreme maximum and minimum temperatures (Figures 9 and 10). The simulation of precipitation in summer is, however, slightly better represented by the NOAH LSM than by the other LSM components (Figure 9). Nonetheless, the comparison of all WRF experiments with observations shows overestimated values of precipitation over most of North America in agreement with other studies using WRF over the

western US (Jin et al., 2010; Chen et al., 2014) and over Europe (Pieri et al., 2015). Atmospheric parameterizations were not tested in our study; however, other WRF sensitivity experiments using several microphysics schemes over Europe found a positive bias in precipitation for all simulations, which was considerably reduced in summer within a convective permitting simulation (Pieri et al., 2015). That is, the positive bias in precipitation has been reported in WRF simulations over different

domains using several LSM components, horizontal resolutions, microphysics parameterizations, and reanalysis products as initial and boundary conditions (Figures 9 in this manuscript, Pieri et al., 2015; Chen et al., 2014; Jin et al., 2010). Therefore, the results included here together with the results reported in the literature suggest that the use of finer resolutions may raise precipitation biases in WRF simulations over North America, but the implementation of convective-permitting processes

and other atmospheric parameterizations could reduce this bias.

## 6    Conclusions

This study has shown the effect of changes in horizontal resolution and LSM choice on the simulation of energy fluxes at the surface and temperature and water conditions in the near-surface. The effect of both model choices intensifies in summer, due to the increase in energy and water ex-

changes between the lower atmosphere and the land surface. Enhancing horizontal resolution leads to higher precipitation climatologies for all LSMs over coastal areas at low latitudes, mainly due to an increase in convective precipitation also associated with an increase in latent heat flux. Our results highlight the important role of the LSM choice in the WRF representation of the energy partition at the surface, which mainly affects the simulation of near-surface temperatures over North Amer-

ica. Additionally, these results demonstrate the impact of the LSM choice on simulated atmospheric



conditions, showing LSM dependent differences in the simulated cloud fraction and non-convective precipitation rates. This is probably associated with land-atmospheric coupled character of our simulations and the interactions between small and large scale dynamical processes.

The evaluation of the WRF simulations against observations supports the use of the CLM4 LSM
as the best choice within the evaluated options for WRF simulations over North America, although it may overestimate temperature extreme events. The use of finer resolutions yield a small improvement in the representation of minimum temperatures within WRF at mid- and high latitudes. Nonetheless, the use of finer resolutions should be implemented with caution since it may increase the WRF bias in mean and extreme precipitation. Further sensitivity experiments using other atmo-
spheric parameterizations or resolutions fine enough for convective-permitting processes are necessary to determine the best WRF configuration for downscaling climate simulations over North America for paleoclimate and climate change studies.

Information provided by downscaling studies are used for building climate change policies, through the information collected in assessment reports (IPCC, 2013; Mbow et al., 2017; Reidmiller et al.,
2019). Thus, sensitivity studies like the one presented here, are crucial to understand and ultimately restrict uncertainties in climate simulations, with direct benefits to society and environment. Particularly, these results should be considered for downscaling studies over North America aimed at projecting future or past conditions and informing policy-makers.

*Code and data availability.* The source code of the Weather Research and Forecasting model (WRF v.3.9) is
http://www2.mmm.ucar.edu/wrf/users/download/get_source.html (access date: August, 2017). The outputs of all simulations together with the code used to estimate the presented results are available at https://doi.org/10.5281/zenodo.5106087. The NARR product was obtained from https://www.ncei.noaa.gov/data/north-american-regional-reanalysis/access/3-hourly/ (access date: July, 2021). The CRU TS4.03 database can be downloaded from the University of East Anglia webpage (https://crudata.uea.ac.uk/cru/data/hrg/, access date: July 2021). The DAYMET V3
database is available at https://daymet.ornl.gov/ (access date: December 2019).

*Author contributions.* AGG designed the modeling experiment, performed the simulations and analyzed model outputs. All authors contributed to the interpretation and discussion of results. AGG wrote the manuscript with continuous feedback from all authors.

*Acknowledgements.* We thank to the Mesoscale and Microscale Meteorology (MMM), the National Center for
atmospheric Research (NCAR), the National Oceanic and Atmospheric Administration (NOAA), the Climatic





Research Unit (CRU) at the University of East Anglia, and the Oak Ridge National Laboratory Distributed Active Archive Center (ORNL DAAC) for making the WRF code and the NARR, CRU and DAYMET datasets available. This analysis contributes to the PALEOLINK project (http://www.pastglobalchanges.org/science/wg/2k-network/projects/paleolink/intro, last access: 4 May 2021), part of the PAGES 2k Network. All WRF sim-

ulations were performed in the computational facilities provided by the Atlantic Computational Excellence Network (ACENET-Compute Canada). This work was supported by grants from the Natural Sciences and Engineering Research Council of Canada Discovery Grant (NSERC DG 140576948), the Canada Research Chairs Program (CRC 230687), and the Canada Foundation for Innovation (CFI) to Hugo Beltrami. During the elaboration of this analysis, Almudena García-García and Francisco José Cuesta-Valero were funded by

Hugo Beltrami's Canada Research Chair program, the School of Graduate Students at Memorial University of Newfoundland and the Research Office at St. Francis Xavier University.

*Competing interests.*   The authors declare that they have no conflict of interest





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

**Table 1.** Summary of the regional simulations performed in this analysis.

| NAME | LSM | Resolution | Vegetation Mode | Simulation Time Step | Radiation Time Step |
|---|---|---|---|---|---|
| NOAH 25 km | | 25 km | Prescribed | 2.5 min | 6 min |
| NOAH 50 km | NOAH | 50 km | Prescribed | 5 min | 20 min |
| NOAH 100 km | | 100 km | Prescribed | 10 min | 20 min |
| NOAH-MP 25 km | | 25 km | Prescribed | 2.5 min | 6 min |
| NOAH-MP 50 km | NOAH-MP | 50 km | Prescribed | 5 min | 20 min |
| NOAH-MP 100 km | | 100 km | Prescribed | 10 min | 20 min |
| NOAH-MP-DV 25 km | | 25 km | Dynamic | 2.5 min | 6 min |
| NOAH-MP-DV 50 km | NOAH-MP | 50 km | Dynamic | 5 min | 20 min |
| NOAH-MP-DV 100 km | | 100 km | Dynamic | 10 min | 20 min |
| CLM4 25 km | | 25 km | Prescribed | 2.5 min | 6 min |
| CLM4 50 km | CLM4 | 50 km | Prescribed | 5 min | 20 min |
| CLM4 100 km | | 100 km | Prescribed | 10 min | 20 min |

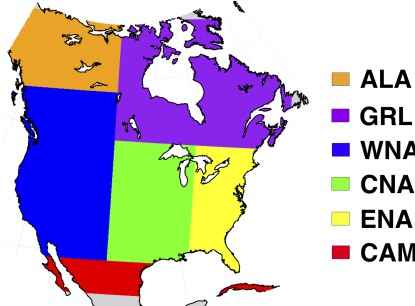

**Figure 1.** Subregions employed for the bias analysis, adapted from Giorgi and Francisco (2000): Central America, CAM; Western North America, WNA; Central North America, CNA; Eastern North America, ENA; Alaska, ALA; and Greenland, GRL.



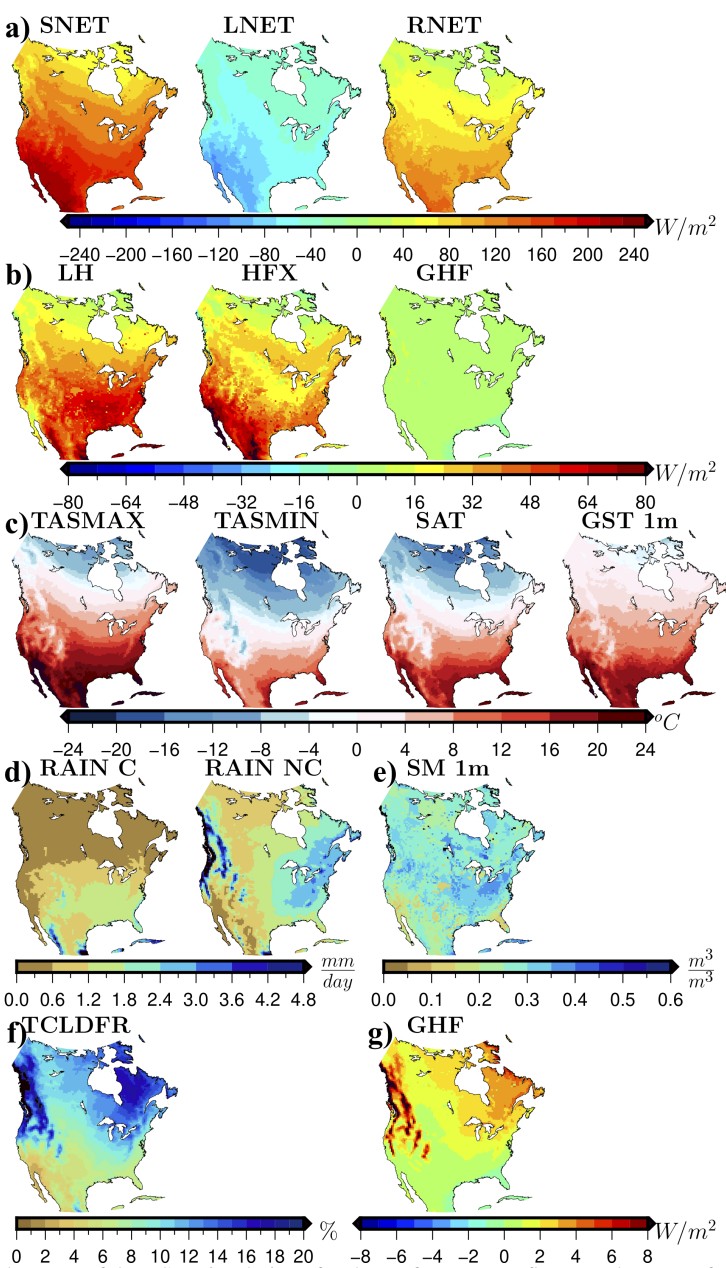

**Figure 2.** Ensemble mean of the LSM simulations for the surface energy fluxes and near-surface conditions (net shortwave radiation SNET; net longwave radiation LNET; surface net radiation RNET (a); latent heat flux LH; sensible heat flux HFX; ground heat flux GHF (b); maximum temperature TASMAX; minimum temperature TASMIN; surface air temperature SAT; soil temperature at 1m depth GST 1m (c); accumulated convective and non convective precipitation at the surface (RAIN C and RAIN NC, d); soil moisture contained in the first soil meter SM 1m (e); and total cloud cover fraction (TCLDFR, f) for the WRF ensemble mean. GHF results are also represented using its own color scale (g). Mean values are estimated as the temporal average for the period 1980-2013 using simulations performed with 50 km resolution.



**Figure 3.** Anomalies of energy fluxes at the surface: a) net shortwave radiation SNET, b) net longwave radiation LNET, c) soil net radiation RNET, d) latent heat flux LH, f) sensible heat flux HFX, and g) ground heat flux GHF for each LSM simulation relative to the LSM ensemble mean. Mean values are estimated as the temporal average for the period 1980-2013 using simulations performed with 50 km resolution. Grid cells with a non-significant change at the 5% significant level are masked in grey.

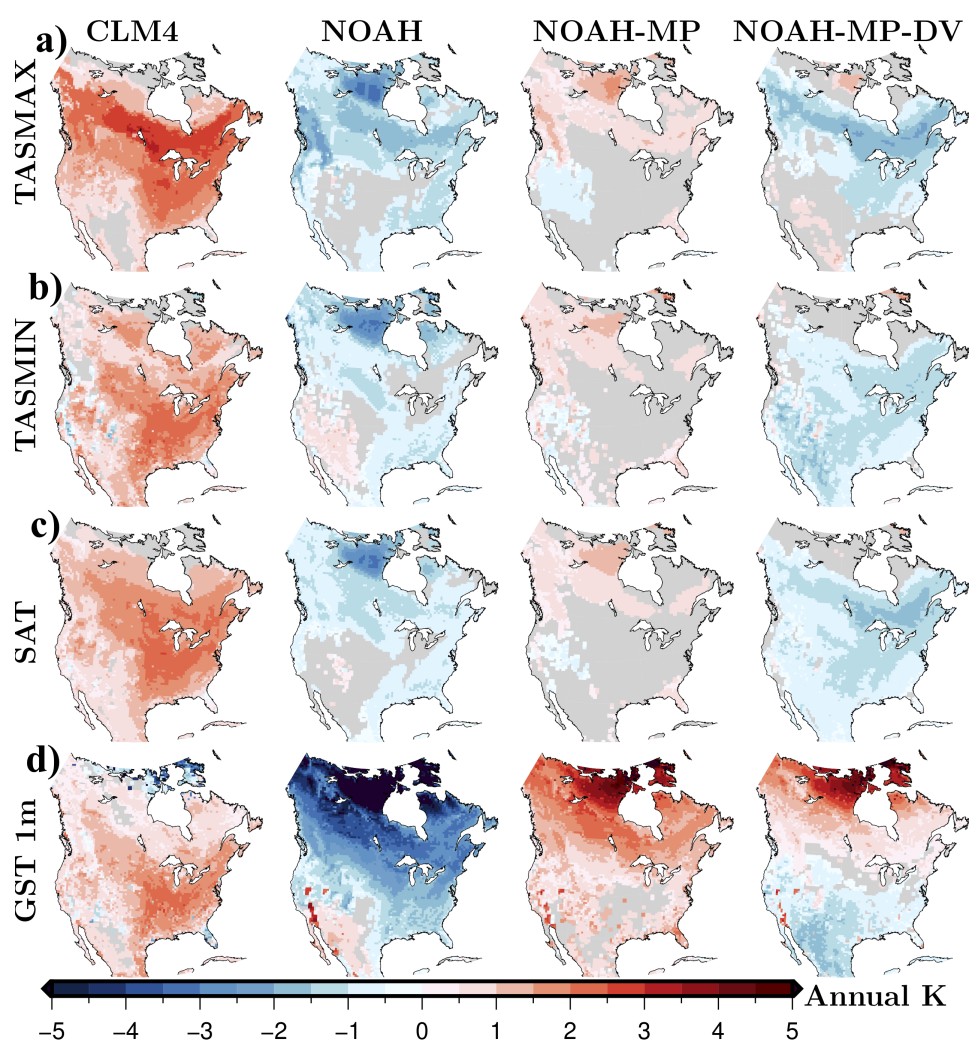

**Figure 4.** Anomalies of near-surface temperature conditions: a) daily maximum temperature TASMAX, b) daily minimum temperature TASMIN, c) surface air temperature SAT, and d) soil temperature at 1m depth GST 1m for each LSM simulation relative to the LSM ensemble mean. Mean values are estimated as the temporal average for the period 1980-2013 using simulations performed with 50 km resolution. Grid cells with a non-significant change at the 5% significant level are masked in grey.

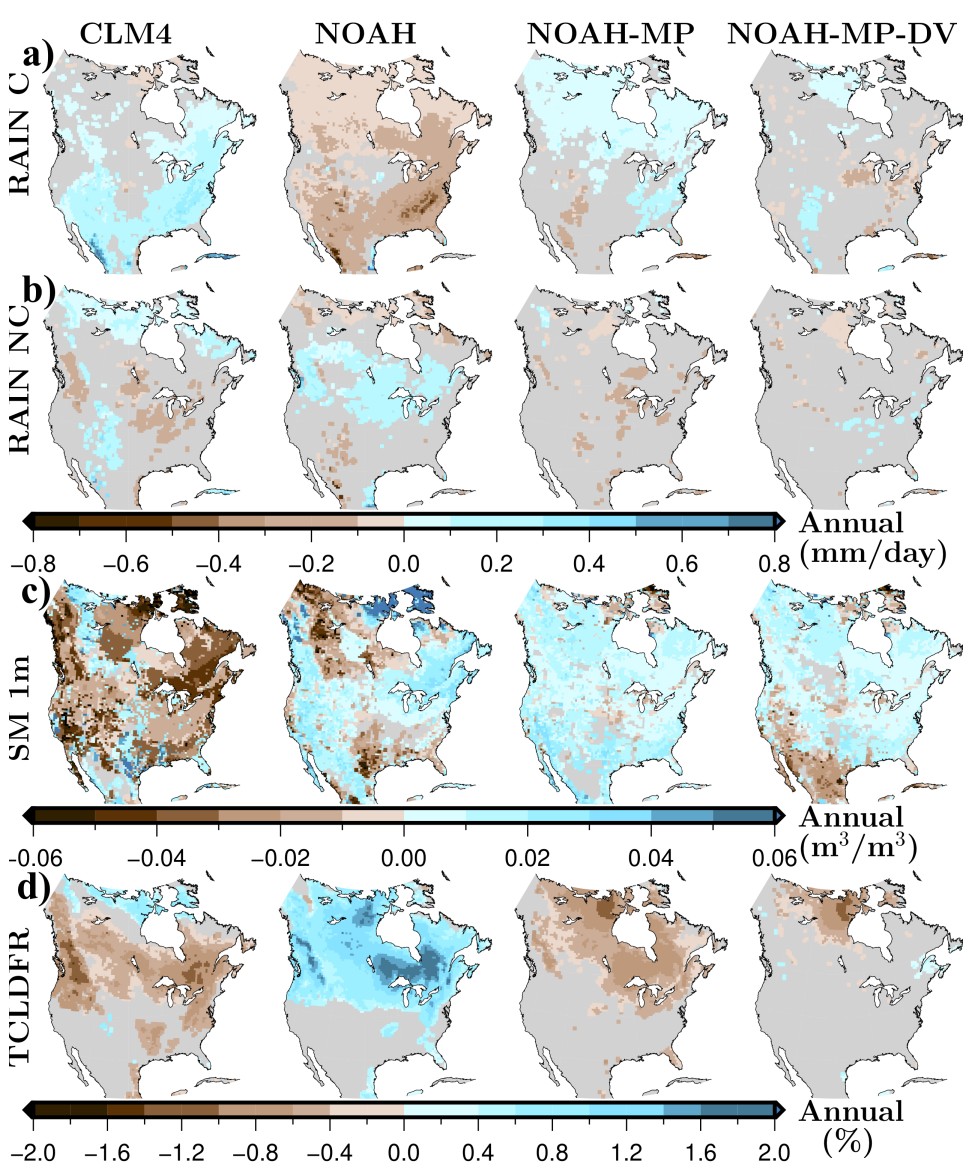

**Figure 5.** Anomalies of near-surface humid conditions: a) accumulated convective and b) non-convective pre-cipitation at the surface RAIN C and RAIN NC, c) soil moisture contained in the first soil meter SM 1m, and d) total cloud fraction TCLDFR for each LSM simulation relative to the LSM ensemble mean. Mean values are estimated as the temporal average for the period 1980-2013 using simulations performed with 50 km resolution. Grid cells with a non-significant change at the 5% significant level are masked in grey.

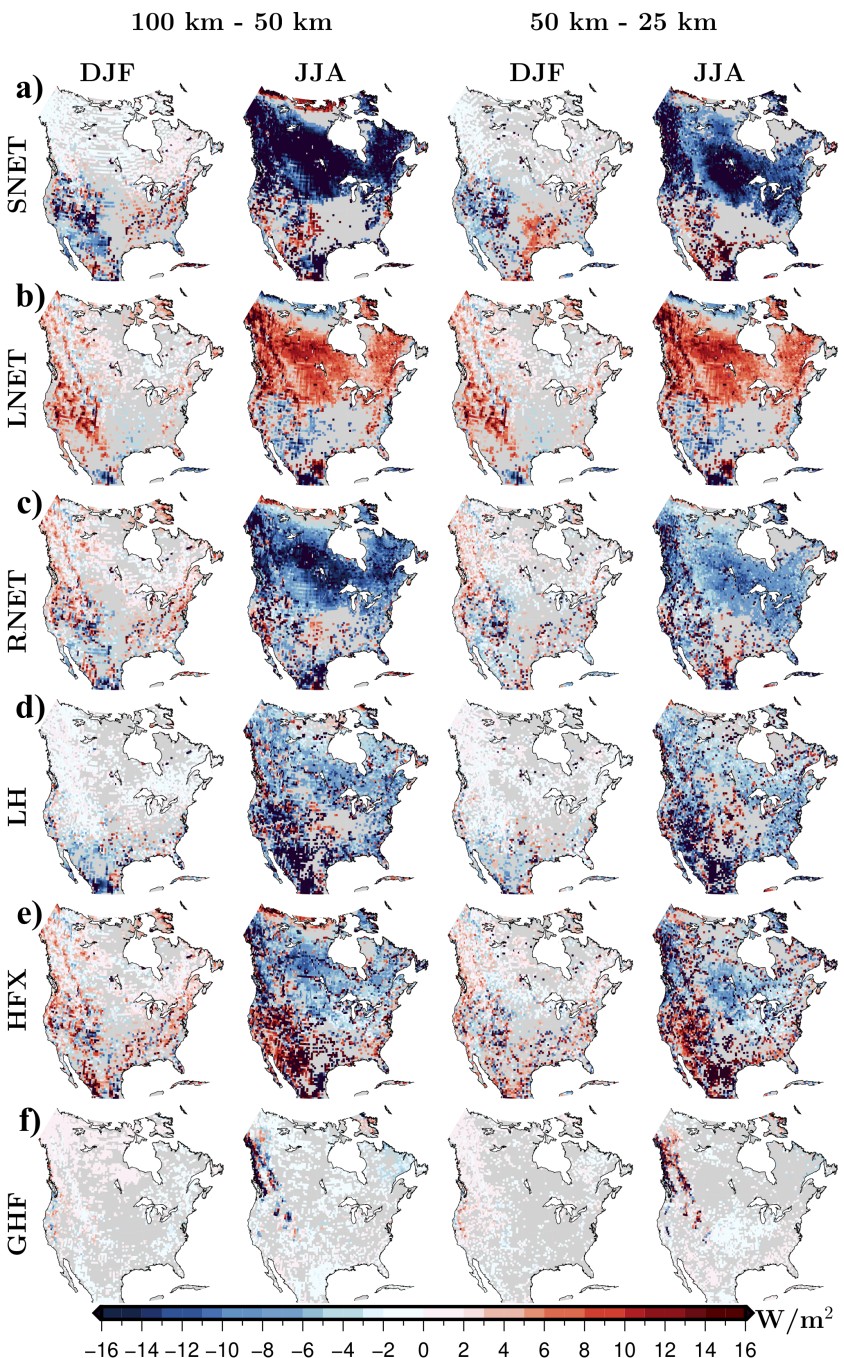

**Figure 6.** Seasonal mean difference in surface energy fluxes between the 100 km and 50 km simulations (left) and between the 50 km and 25 km simulations (right). All outputs are from the NOAH-MP-DV simulations for the period 1980-2013. Grid cells with a non-significant change at the 5% significant level are masked in grey. All outputs from the 25, 50 and 100 km simulations were mapped to a common grid (CRU grid) using the nearest model grid point.



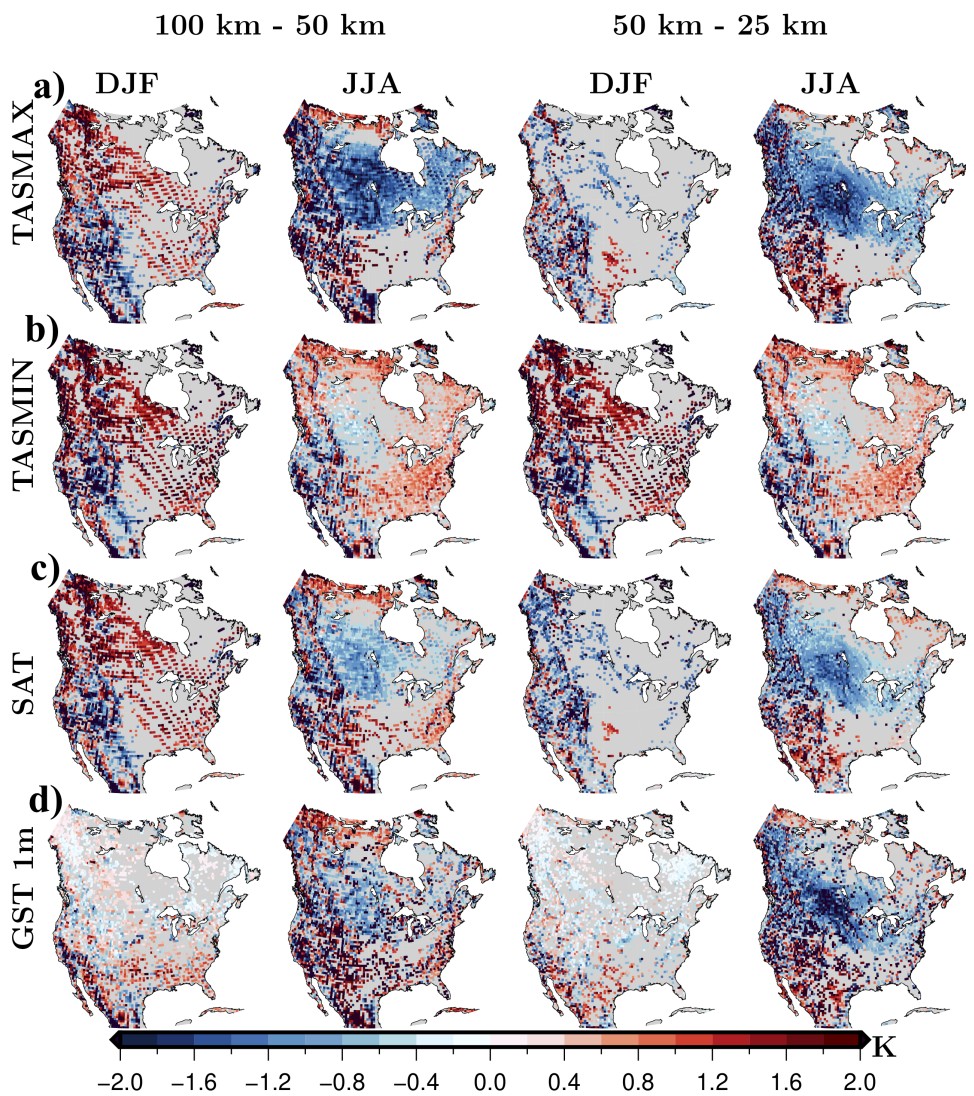

**Figure 7.** Seasonal mean difference in near-surface temperature conditions between the 100 km and 50 km simulations (left) and between the 50 km and 25 km simulations (right). All outputs are from the NOAH-MP-DV simulations for the period 1980-2013. Grid cells with a non-significant change at the 5% significant level are masked in grey. All outputs from the 25, 50 and 100 km simulations were mapped to a common grid (CRU grid) using the nearest model grid point.

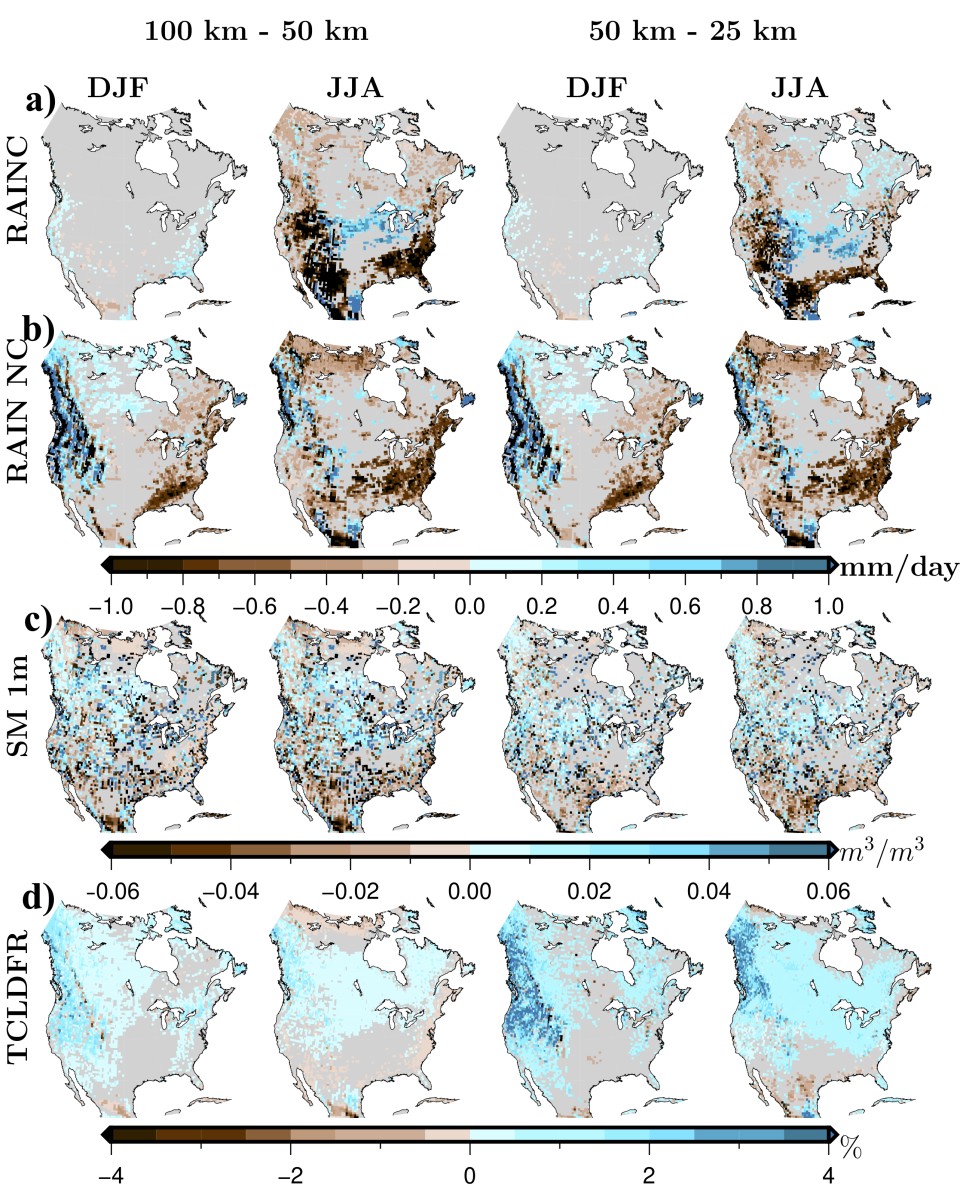

**Figure 8.** Seasonal mean difference in near-surface humid conditions between the 100 km and 50 km simulations (left) and between the 50 km and 25 km simulations (right). All outputs are from the NOAH-MP-DV simulations for the period 1980-2013. Grid cells with a non-significant change at the 5% significant level are masked in grey. All outputs from the 25, 50 and 100 km simulations were mapped to a common grid (CRU grid) using the nearest model grid point.





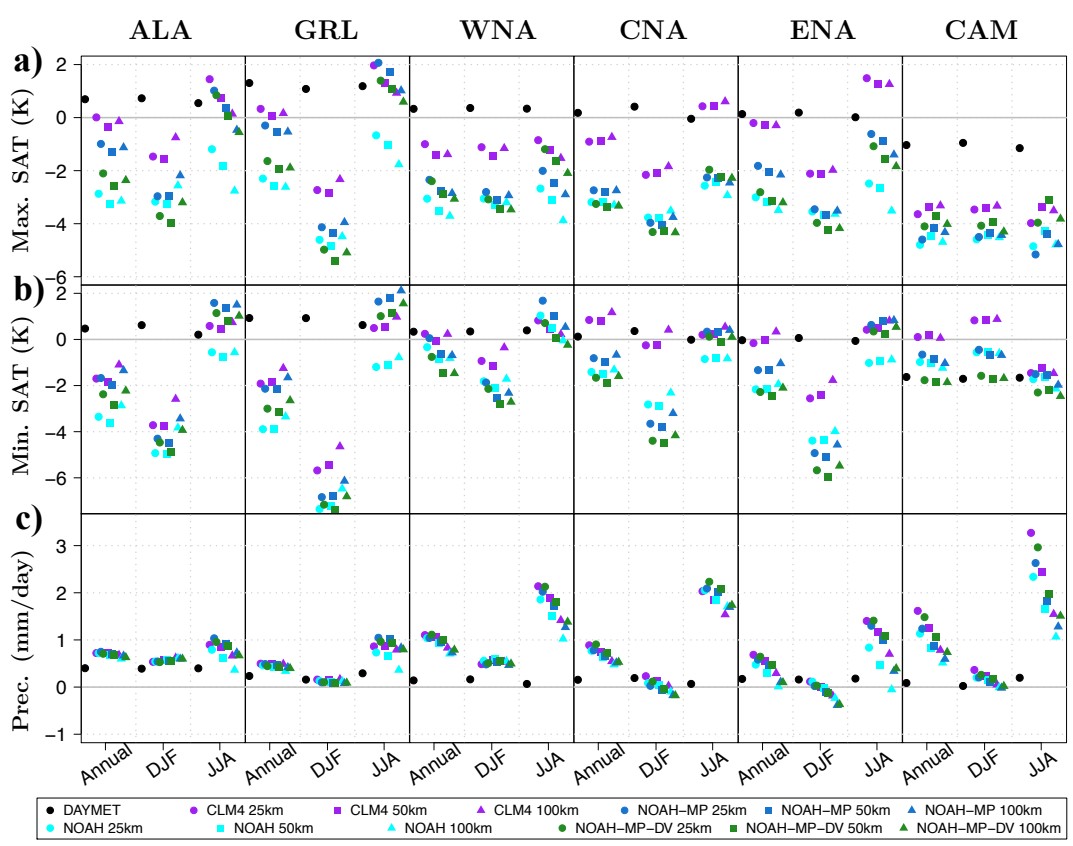

**Figure 9.** Regional mean annual and seasonal bias in maximum and minimum temperature ($^oC$) and in accumulated precipitation (mm/day) for all experiments and the DAYMET data product relative to the CRU database from 1980 to 2013. Biases are estimated averaging over six subregions (Figure 1) adapted from Giorgi and Francisco (2000): Central America, CAM; Western North America, WNA; Central North America, CNA; Eastern North America, ENA; Alaska, ALA; and Greenland, GRL.



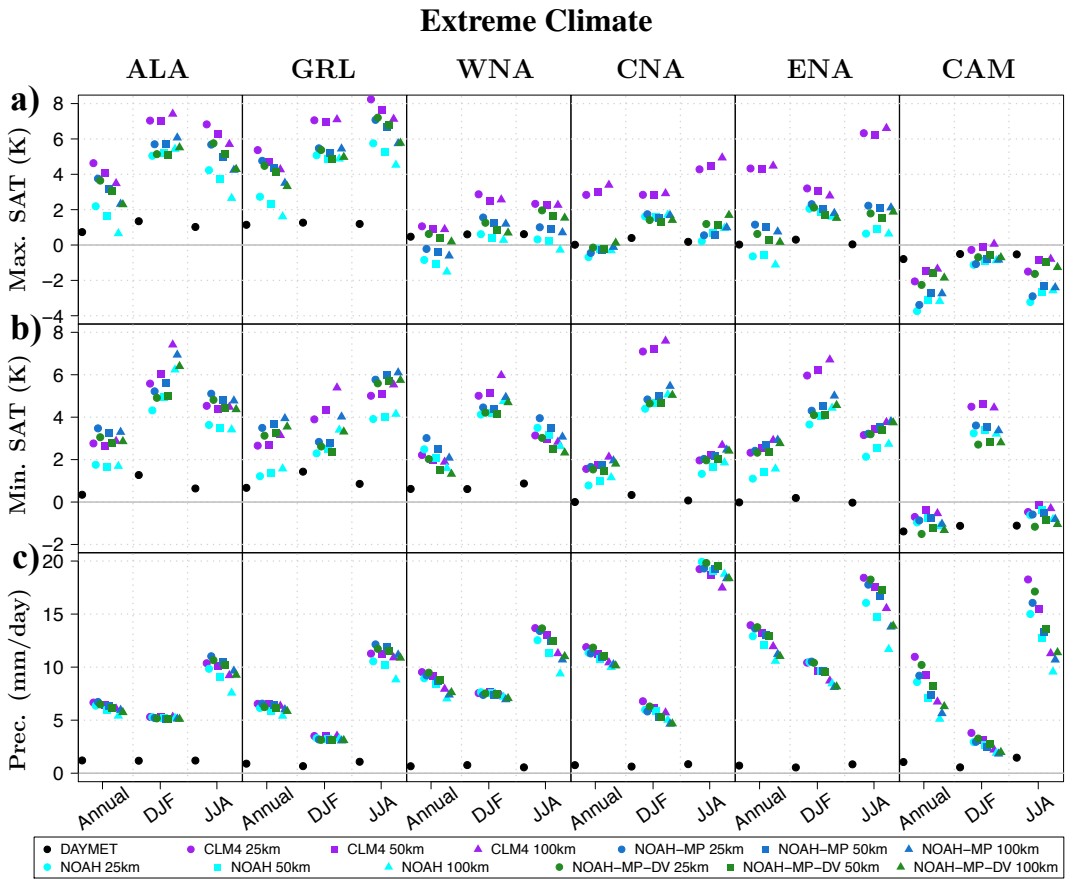

**Figure 10.** Regional mean annual and seasonal bias in extreme maximum and minimum temperature ($^oC$) and in extreme precipitation (mm/day) for all experiments and the DAYMET data product relative to the CRU database from 1980 to 2013. Extremes are calculated as the 95th percentile of the annual and seasonal temporal series at each grid cell. Biases are estimated averaging over six subregions (Figure 1) adapted from Giorgi and Francisco (2000): Central America, CAM; Western North America, WNA; Central North America, CNA; Eastern North America, ENA; Alaska, ALA; and Greenland, GRL.