# Peer review of "WRF v.3.9 sensitivity to land surface model and horizontal resolution changes over North America"

_Geoscientific Model Development, 2021_

## Referee Comment (RC2)

The authors designed multiple WRF experiments to evaluate and compare the influence of the LSM choice and horizontal resolution, on the energy and water fluxes at the surface and near-surface conditions over North America. This is a very important work as large-scale models go to finer spatial resolution with the advances in computational resources and high-resolution data availability. Also, understanding the advantage and disadvantages of different land surface models (LSMs) with different process parameterization is crucial to understand and restrict uncertainties in climate simulations. Overall, the paper is well written and within the scope of GMD. I recommend accepting this paper with a moderate revision.

**Specific comments:**

- L41. Besides the soil physics, other land surface processes (e.g., vegetation, groundwater) could also affect the land–atmosphere interactions. Instead of only mentioning the soil physics here, you should also mention other vital processes. As you also concluded in L280, "This suggests that the different representation of vegetation in each LSM yields to different estimates of soil properties." Please summarize more about the difference in LSMs here. Otherwise, the reader may think soil physics is the most critical determination reason for LSMs. In L430, it seems that you prefer to refer LSMs as soil schemes, which is kind of too simplified.

- The way you are explaining different results between LSMs is vague, e.g., the paragraph around L215. Which are possible major differences between LSMs cause these different simulations is not well explained. It is beneficial, but it may not be easy, to provide more information and commentary/insights, which should be very useful to guide the LSMs' development in the future.

- In section 4.2, you analyzed the difference caused by different resolutions. You explained the difference in simulated variables by using other simulated variables (e.g., its components). For example, L295, net shortwave radiation -> net total radiation. These explanations are indeed needed. However, it is not clear why net shortwave radiation is changed due to finer resolution. It is helpful to explain, from more bottom processes, how

resolution increase changes the energy or water simulations (e.g., finer resolution of DEM or LULC, and how).

- You compared 3 different resolutions (i.e.,25, 50, 100 km) in this paper. As large-scale modeling goes higher resolution or Hyperresolution, for example, NLDAS using 12.5 km, or 1 km (e.g., wood et al., 2011), it would be helpful to provide more comments on this in the discussion.

**Reference**

Wood, Eric F., et al. "Hyperresolution global land surface modeling: Meeting a grand challenge for monitoring Earth's terrestrial water." Water Resources Research 47.5 (2011). https://agupubs.onlinelibrary.wiley.com/doi/full/10.1029/2010WR010090

---

## Author Response (AR1)

**Response to Reviewers Document for GMD-2021-243 by Almudena García-García, Francisco José Cuesta-Valero, Hugo Beltrami, Fidel González-Rouco and Elena García-Bustamante.**

**We are grateful for the thoughtful and constructive feedback of both reviewers.**

**This Response to the Reviewers document provides a complete description of the changes that have been made in response to each individual reviewer comment. Reviewer comments are shown in plain text. Author responses are shown in bold blue text. All line numbers in the author responses refer to locations in the revised manuscript with changes marked.**

**Referee #1**

This manuscript explores the effect of horizontal resolution and land surface model (LSM) choice on the simulation of WRF surface energy fluxes and conditions. This is a very well written paper on a topic that is not a focus for WRF. Most WRF sensitivity tests deal with the various combinations of the atmospheric parameterizations. I like that they analyze not just mean air temperature and precipitation but also minimum and maximum temperature and convective and non-convective precipitation as well. The authors find that CLM4 is the best LSM to use with WRF. They find that model horizontal resolution most affects precipitation.

I think the description of the WRF sensitivity tests in this manuscript is a perfect fit for GMD. Practically speaking, this is very helpful for WRF users who want to run regional climate simulations. Scientifically, it lends no information to what aspects of the LSMs beyond just simply model complexity causes the improved simulations with CLM4, but that would require a deeper dive into offline LSM simulations that would go beyond the scope of GMD. It is interesting that the authors suggest that further sensitivity tests should be performed at convective-permitting resolutions. What would be the horizontal resolution for that? Are such resolutions computationally achievable for continental-scale simulations as is done here at this time?

**We thank the reviewer for the positive feedback. A convective-permitting simulation would require much finer horizontal resolutions (< 5 km). Indeed, climate simulations at continental scales would require large computational resources at those resolutions. For example, around 1000 Tb would be required just to store the outputs of four WRF experiments of 30 years over North America with a resolution of 5 km. Hence a reduction in the area of interest or in the period of the simulation may be necessary to perform a sensitivity analysis including convective permitting simulations. We have included a small discussion on this topic in the new version of the manuscript (see lines 492-498).**

Additionally, I would suggest one minor change to the manuscript: The authors use the abbreviation RAIN for total precipitation (rainfall + snowfall). I would suggest using PRECIP (or something similar) instead to avoid confusion for the casual reader.

**We agree with the reviewer that the term "RAIN" can be confusing, so we have changed the text and the figures accordingly.**

**Referee #2**

The authors designed multiple WRF experiments to evaluate and compare the influence of the LSM choice and horizontal resolution, on the energy and water fluxes at the surface and near-surface conditions over North America. This is a very important work as large-scale models go to finer spatial resolution with the advances in computational resources and high-resolution data availability. Also, understanding the advantage and disadvantages of different land surface models (LSMs) with different process parameterization is crucial to understand and restrict uncertainties in climate simulations. Overall, the paper is well written and within the scope of GMD. I recommend accepting this paper with a moderate revision.

**We thank the reviewer for the positive feedback.**

Specific comments:

• L41. Besides the soil physics, other land surface processes (e.g., vegetation, groundwater) could also affect the land–atmosphere interactions. Instead of only mentioning the soil physics here, you should also mention other vital processes. As you also concluded in L280, "This suggests that the different representation of vegetation in each LSM yields to different estimates of soil properties." Please summarize more about the difference in LSMs here. Otherwise, the reader may think soil physics is the most critical determination reason for LSMs.

**Indeed, the representation of land cover type, soil water content, snow cover, drip, runoff or infiltration are example of processes that might strongly affect the simulation of land-atmosphere interactions. We have modified the introduction according to the reviewer's comment (see line 45).**

In L430, it seems that you prefer to refer LSMs as soil schemes, which is kind of too simplified.

**Agreed. We have replaced "soil scheme" by soil model or LSM in the new version of the manuscript (see for example lines 368 and 453).**

• The way you are explaining different results between LSMs is vague, e.g., the paragraph around L215. Which are possible major differences between LSMs cause these different simulations is not well explained. It is beneficial, but it may not be easy, to provide more information and commentary/insights, which should be very useful to guide the LSMs' development in the future.

**As the reviewer pointed out it could be highly beneficial but also vey difficult to find the LSM structural differences that lead to our results. For the aim of finding those features in LSMs that lead to large LSM differences in the representation of surface fluxes, we selected the set of simulations including a rather basic LSM (NOAH), the upgrade of that LSM (NOAH-MP), and the most complex LSM available in WRF (CLM4). Additionally, two simulations with the same LSM using prescribed and dynamic vegetation were included in the ensemble to test the role of dynamic vegetation in the simulation of land-atmosphere interactions. However, our analysis**

reveals large differences among LSM outputs over the whole domain, particularly large between the NOAH and the CLM4 LSMs and over highly vegetated areas but also over dry areas. Since it is difficult to find the features in the LSM codes leading to these results, we try to relate LSMs differences between variables. Thus, we can conclude that the LSM differences in SNET radiation are probably related to the different albedo values calculated within each LSM, the large LSM differences in sensible heat flux over the boreal forest are likely related to the LSM estimates of surface roughness, while the latent heat flux differences are probably associated with different estimates of evaporative resistances and the treatment of soil water in each LSM. However, all of these "causes" are hypotheses and identifying the role (not just the value) of each of these properties in each LSM would require very specific analyses that are by far out of the scope of the present work. We have included a few sentences about these hypotheses in the discussion (see lines 433-439).

• In section 4.2, you analyzed the difference caused by different resolutions. You explained the difference in simulated variables by using other simulated variables (e.g., its components). For example, L295, net shortwave radiation -> net total radiation. These explanations are indeed needed. However, it is not clear why net shortwave radiation is changed due to finer resolution. It is helpful to explain, from more bottom processes, how resolution increase changes the energy or water simulations (e.g., finer resolution of DEM or LULC, and how).

Changes in downward shortwave radiation at the surface are mainly driven by changes in cloud cover, but changes in atmospheric water vapor and aerosols may also affect the shortwave radiation reaching the ground surface (Hatzianastassiou et al., 2005). Changes in surface albedo also lead to changes in the upward component of shortwave radiation, thus affecting net shortwave radiation (SNET). The resolution differences in estimating albedo and land-use/land-cover in our results are smaller than those associated with the cloud cover and the microphysics of the model. This can be seen in Figure 1 in this document, that represents the resolution effect on both components of net shortwave radiation. The effect of resolution on the downward component of shortwave radiation, which is dependent on cloud formation and atmospheric composition, is larger than the resolution effect on the upward component of shortwave radiation, which is dependent on albedo values. Nevertheless, the resolution changes in winter SNET at southern latitudes of the Rocky Mountains that are not related to cloud cover seem to be associated with the upward component of shortwave radiation and therefore with the resolution effect on surface albedo values (Figure 6a and 8d in the manuscript).

The changes in cloud cover with resolution may be caused by the performance of the microphysical parameterizations at different resolutions and the improvement in the representation of orography (Pieri et al 2015 and Prein et al., 2016).

We have included this discussion and Figure 1 in the new version of the manuscript (see lines 302-314, 449-452 and Figure S14).

[Figure]

*Figure 1: Figure 1: Seasonal mean difference in downward, upward and net shortwave radiation at the surface (SWDNB, SWUPB, SNET) between the 100 km and 50 km simulations (left) and between the 50 km and 25 km simulations (right). All outputs are from the NOAH-MP-DV simulations for the period 1980-2013. Grid cells with a non-significant change at the 5% significant level are masked in grey. All outputs from the 25, 50 and 100 km simulations were mapped to a common grid (CRU grid) using the nearest model grid point.*

- You compared 3 different resolutions (i.e.,25, 50, 100 km) in this paper. As large-scale modeling goes higher resolution or Hyperresolution, for example, NLDAS using 12.5 km,

or 1 km (e.g., wood et al., 2011), it would be helpful to provide more comments on this in the discussion.

**The NLDAS products and the Hyperresolution land surface modeling suggested in Wood et al., 2011 propose the use of "uncoupled" Land Surface Models (LSM) at very high resolution. That is, the LSM uses atmospheric forcings to run the simulation, but the evolution of soil and surface conditions do not modify atmospheric conditions, removing the feedback between soil and atmosphere. The role of surface soil or land cover properties in the simulation of surface fluxes and near-surface conditions changes largely when using "coupled" or "uncoupled" LSMs (Lague et al., 2019). In this sensitivity analysis, we used coupled land-atmosphere simulations and therefore it is difficult to extrapolate our conclusions to "uncoupled" LSM simulations at higher resolutions.**

**Changes in the WRF simulations from 100 km to 50 km and from 50 km to 25 km have a similar behaviour. Thus, considering coupled land-atmosphere simulations, the use of resolutions finer than 25 km in our analysis will probably lead to similar conclusions, unless there is a change in the hydrostatic balance of the model, that may happen at around 10 km, or the direct representation of convective processes at resolutions lower than 5 km. We have included more information about this in the new version of the manuscript (see lines 492-496).**

Reference
Wood, Eric F., et al. "Hyperresolution global land surface modeling: Meeting a grand challenge for monitoring Earth's terrestrial water." Water Resources Research 47.5 (2011). https://agupubs.onlinelibrary.wiley.com/doi/full/10.1029/2010WR010090

**References**

**Hatzianastassiou, N., Matsoukas, C., Fotiadi, A., Pavlakis, K. G., Drakakis, E., Hatzidimitriou, D., and Vardavas, I.: Global distribution of Earth's surface shortwave radiation budget, Atmos. Chem. Phys., 5, 2847–2867, https://doi.org/10.5194/acp-5-2847-2005, 2005.**

**Laguë, M. M., Bonan, G. B., and Swann, A. L. S.: Separating the impact of individual land surface properties on the terrestrial surface energy budget in both the coupled and un-coupled land-atmosphere system, Journal of Climate, 32, 5725–5744, https://journals.ametsoc.org/doi/abs/10.1175/JCLI-D-18-0812.1, 2019.**

**Pieri, A. B., von Hardenberg, J., Parodi, A., & Provenzale, A. (2015). Sensitivity of Precipitation Statistics to Resolution, Microphysics, and Convective Parameterization: A Case Study with the High-Resolution WRF Climate Model over Europe, Journal of Hydrometeorology, 16(4), 1857-1872. Retrieved Nov 14, 2021, from https://journals.ametsoc.org/view/journals/hydr/16/4/jhm-d-14-0221_1.xml**

**Prein, A. F., et al. "Precipitation in the EURO-CORDEX 0.11º and 0.44º simulations: high resolution, high benefits?." *Climate dynamics* 46.1-2 (2016): 383-412.**